# Covalent Crosslinking Cellulose/Graphene Aerogels with High Elasticity and Adsorbability for Heavy Metal Ions Adsorption

**DOI:** 10.3390/polym15112434

**Published:** 2023-05-24

**Authors:** Peipei Sun, Meng Wang, Tingting Wu, Longsuo Guo, Wenjia Han

**Affiliations:** State Key Laboratory of Biobased Material and Green Papermaking, Advanced Materials Institute, Qilu University of Technology, Shandong Academy of Sciences, Jinan 250353, China; sunpeipei@sdas.org (P.S.); hwj200506@163.com (W.H.)

**Keywords:** cellulose nanofibers, aerogel, adsorption, heavy metal

## Abstract

With the fast development of modern industry, heavy metal contaminant became more severe. How to remove heavy metal ions in water in a green and efficient way is a prominent problem in current environmental protection. The adsorption of cellulose aerogel as a novel heavy metal removal technology has many advantages, including abundant resources, environmental friendly, high specific surface, high porosities and without second pollution, which means it has a wide application prospect. Here, we reported a self-assembly and covalent crosslinking strategy to prepare elastic and porous cellulose aerogels using PVA and graphene and cellulose as precursor. The resulting cellulose aerogel had a low density of 12.31 mg cm^−3^ and excellent mechanical properties, which can recover to its initial form at 80% compressive strain. Meanwhile, the cellulose aerogel had strong adsorption capacity of Cu^2+^ (80.12 mg g^−1^), C_d_^2+^ (102.23 mg g^−1^), Cr^3+^ (123.02 mg g^−1^), Co^2+^ (62.38 mg g^−1^), Zn^2+^ (69.55 mg g^−1^), and Pb^2+^ (57.16 mg g^−1^). In addition, the adsorption mechanism of the cellulose aerogel was investigated using adsorption kinetics and adsorption isotherm, and the conclusion was that the adsorption process was mainly controlled by chemisorption mechanism. Therefore, cellulose aerogel, as a kind of green adsorption material, has a very high application potential in future water treatment applications.

## 1. Introduction

The pollution to the environment and the harm to human health of heavy metal ions was found to be increasingly serious. With the rapid development of industry and the acceleration of urbanization, more and more industrial and mining wastewater and domestic sewage are directly discharged without proper treatment, causing heavy metal pollution in water. Heavy metal ions not only pollute water, but also accumulate in the environment and organisms, and even trace amounts can cause various diseases and biological variations [1]. At present, the commonly used methods to remove heavy metal ions from water include chemical coagulation, ion exchange, membrane separation, electrochemical technology, ultrafiltration, and so on [2]. Nevertheless, the disadvantages of these technologies are low efficiency, complex process, expensive, and ease to produce other pollutants. The research showed that adsorption is considered to be the most effective method to deal with a large amount of water pollution [3,4]. The adsorption method refers to making use of porous solid substances to adsorb heavy metal ions in water to achieve the purpose of treating, reducing and even eliminating heavy metal pollution. It has the advantages of high efficiency, fast speed, low cost, and wide applicability. Therefore, it is urgent for scientific researchers to develop high-performance, low-cost, and environmentally friendly adsorbents.

Cellulose is one of the most abundant natural biomass material and mainly derived from the cells of green plants; it is the most widely distributed, abundant, and renewable natural macromolecular material in nature. The yield of cellulose through photosynthesis is up to 150 billion tons per year in the world, which is an inexhaustible renewable resource [5]. Cellulose aerogel is a porous solid obtained by means of dissolution or dispersion of cellulose as precursor or by special drying methods, including freeze drying and supercritical drying. As a kind of porous material, cellulose aerogel has a high specific surface area and abundant pore structure, which has very high application value in the field of adsorption [6]. Cellulose aerogels rapidly became a new hotspot in the field of aerogels because of their maintaining the basic structural characteristics of traditional aerogels and also incorporating the advantages of green and renewable raw materials, which are biodegradable, flexible, and easy to process [7,8,9]. Tan et al. successfully prepared cellulose derivative aerogel by chemical crosslinking on cellulose ester [10]. However, the weak hydrogen bond between cellulose leads to poor mechanical properties of cellulose aerogel, which greatly limits its practical application. How to achieve cellulose aerogel with high efficiency adsorption capacity and marvelous mechanical properties is still one of the most difficult problems.

As a natural polymer aerogel, it can not only be used as a carrier of active substances or raw materials but also as a template to further prepare composite aerogels with other functional properties. Hydrogen bonding between molecules is one of the bases for the formation of certain physical structures and mechanical strength of cellulose aerogels, but this bonding will also lead to the contraction and aggregation of cellulose matrix, resulting in the collapse of pore structure in gel, affecting the adsorption capacity and mechanical strength of cellulose aerogels and limiting the further application [11]. Single-layer inorganic nano-sheets were the focus of research in the field of nanocomposites due to their surface effect, volume effect, and quantum size effect [12]. Recently, some studies combined two-dimensional materials including clay, graphene, and MXene with cellulose to improve the properties of cellulose aerogel. Zhao et al. [13] reported a silicified BC/MXene aerogel preparing from bacterial cellulose (BC) treated with methyl trimethoxysilane (MTMS) and MXene suspension by freeze-drying, showing outstanding elasticity and specific electromagnetic interference shielding effectiveness. The introduction of inorganic nanosheets with outstanding performance into cellulose aerogels is the key to prepare cellulose composite aerogels with excellent mechanical properties and high adsorption performance [14,15].

Graphene is a kind of honeycomb two-dimensional (2D) nano-sheet formed by closely stacking single-layer sp^2^ hybrid carbon atoms through covalent bonds. It has excellent mechanical, electrical, optical, thermal, and other properties, and it is a new type of carbon material with great development potential. It is reported that the Young’s modulus of single-layer graphene reaches 1 TPa, its breaking strength is high (~125 GPa), and its specific surface area is 2630 m^2^ g^−1^ [16]. Due to its high specific surface area and mechanical properties, graphene shows very high application potential in removing heavy metals and organic pollutants from water [17,18,19]. As a derivative of graphene, graphene oxide has rich oxygen-containing groups, good chemical modifiability and hydrophilicity, and is also widely used in adsorption. Zhang et al. [20] used environmentally friendly cellulose solvent (NaOH/thiourea/aqueous solution) to dissolve cellulose by precooling at low temperature, and then, mixing with GO to prepare cellulose/GO aerogel at room temperature. The research showed that cellulose and GO were enveloped by a strong hydrogen bond, GO promoted the sol gel reaction of cellulose and acted as a cross-linking agent. The Young’s modulus and compressive strength of the composite aerogel increased by 90% and 30% by adding GO, respectively. Therefore, graphene and graphene oxide are ideal inorganic nanosheets to enhance adsorption in cellulose aerogels. 

Therefore, in this study, we report a simple and efficient technique to prepare high performance nanocomposite aerogel by super-assembly of cellulose (C), polyvinyl alcohol (PVA), and graphene (G). This composite aerogel can not only make full use of the physical and chemical properties of single-layer graphene, solve the problem of easy agglomeration between graphene sheets, but also give the composite aerogels with uniform and dense porosity, which can effectively improve the adsorption efficiency. By studying the chemical structure, mechanical properties and adsorption properties of the aerogel, the cross-linking mechanism and adsorption mechanism of aerogel were deduced. We also explored the relationship between the structure of composite aerogel and adsorption. The composite aerogel is a promising adsorption material in the field of heavy metal ion removal.

## 2. Experimental and Materials

### 2.1. Materials

The cellulose raw material (pulp) was provided by Sun paper industry (Jining, China). Concentrated sulfuric acid (98%), CuSO_4_·5H_2_O was purchased from Beijing Chemical Plant; CdCl_2_·2.5H_2_O, CrCl_3_·6H_2_O, CoCl_2_, ZnCl_2_, and PbCl_2_^+^ were purchased from Chinese Academy of Metrology (Beijing, China). The 2,2,6,6-tetramethylpiper-idyl-1-oxyl (TEMPO), sodium hypochlorite solution (NaClO, 8%), sulfuric acid (H_2_SO_4_), phosphoric acid (H_3_PO_4_, 85 wt%), sodium hydroxide (NaOH), sodium nitrate (NaNO_3_), sodium bromide (NaBr), hydrochloric acid (HCl), and potassium permanganate (KMnO_4_), polyvinyl alcohol (PVA, Mw = 88,000), and citric acid (C_6_H_8_O_7_) were supplied by Macklin Industrial Corporation (Shanghai, China). Ultrapure water (18.2 MV/cm) was used in used in all solution configurations.

### 2.2. Characterization

The characterization of the sample was as follows: the micromorphology of nano-cellulose and graphene were measured by transmission electron microscope (TEM, H-7650, Hitachi, Tokyo, Japan). The sample was first dispersed into an ethanol solution and ultrasonic at 800 w for 20 min, and was then dropped onto the surface of the copper for drying and TEM test. The morphologies of the composite aerogel were observed using Hitachi S4800 instrument operated at 2 kV for gold-sputtered samples. The SEM samples were coated with gold for 60 s. Fourier transform infrared spectroscopy (FTIR, Thero Electron, Waltham, MA, USA) measurements were tested on a NICOLET 5700. X-ray diffraction patterns (XRD, D8 ADVANCE, Bruker, Saarbrucken, Germany) were recorded with a Cu Kα radiation at 36 kV and 20 mA. The mechanical property of CPGA was measured using a Universal mechanical tester (TAXT PLUS, Stable Micro System, London, Britain). The sample was compressed at a constant rate on the test table. TGA was carried out using a thermo-analysis instrument (TGA-Q50 type, TA Company, Newcastle, USA) from 25 °C to 600 °C at a heating rate of 10 °C/min under N_2_ atmosphere. The concentration of heavy metal ions in the supernatant was determined by inductively coupled plasma atomic emission spectrometry ICP-MS (ICP-RQ, Thermo Fisher, Waltham, MA, USA).

### 2.3. Preparation of Cellulose/PVA/Graphene Composite Aerogel

The TEMPO-oxide cellulose (TOC) was prepared as follows: the cellulose pulp (2 g), NaBr (0.2 g), and TEMPO (0.0032 g) were added to water (200 mL) under stirring at 25 °C. Then, sodium hypochlorite was added to start the oxidation reaction, and the PH of the solution would be reduced during the oxidation process. After adding the NaOH solution (0.5 mol L^−1^) to maintain the PH of the solution at 10. After 10 h, 2 mL glycol was added to stop the reaction, and then, the cellulose was repeatedly filtered and washed to neutral. Cellulose was homogenized 5 times in a high pressure homogenizer at 100 MPa to obtain nanocellulose dispersion. The content of carboxyl group in TOC measured by conductivity was 0.05 mmol g^−1^. The TOC dispersion was adjusted to 1 wt% by adding ultrapure water. 

To prepare the TOC/PVA/Graphene aerogel (CPGA), PVA (1 g) was added to the TOC suspensions (1 wt%, 100 mL) with the constantly stirring at 95 °C for 2 h. Then, H_3_PO_4_ (2 mL), C_6_H_8_O_7_ (2 g), and graphene (0.02 g) were added to the above solution with stirring for 30 min. The solution was then poured into plastic molds and frozen in an ultra-low temperature refrigerator (−70 °C, 5 h). The aerogel was frozen and freeze-dried in a freeze-dryer (0.001 MPa, 48 h). Eventually, the CPGA was obtained by heating at 60 °C for 30 min in a dryer until completed its esterification crosslinking. As a control, TOC aerogel (CA) was prepared by directly freeze-dried TOC suspensions (1.0 wt%) and TOC/PVA aerogel (CPA) was prepared without graphene during the preparation of CPA. 

### 2.4. Prepare of Heavy Metal Ion Solution

A 500 mg/L Cu^2+^ solution was obtained by dissolving 1.9644 g of CuSO_4_·5H_2_O solid in 1000 mL of deionized water. Other different concentrations can be adjusted by dilution. The C_d_^2+^, Cr^3+^, Co^2+^, Zn^2+^, and Pb^2+^ solutions with different concentrations were prepared in the same way. Metal element (Cd^2+^, Cr^3+^, Co^2+^, Zn^2+^, and Pb^2+^ solution) standard stock solution were prepared by CdCl_2_·2.5H_2_O, CrCl_3_·6H_2_O, CoCl_2_, ZnCl_2_, and PbCl_2_, respectively.

### 2.5. Adsorption Process

A certain mass of freeze-dried cellulose/graphene composite aerogel was placed in 50 mL 200 mg/L Cu^2+^ solution, and was shaken for 120 min at the rate of 100 rpm. After the solution reached the adsorption equilibrium, the aerogel and the adsorption solution were separated directly. Then, the adsorption solution was filtered and collected by a 0.22 um needle filter. The residual concentration of Cu^2+^ in the filtrate was analyzed by ICP-MS. We then took the average of three adsorption experiments. 

The adsorption capacity of metal ions per unit adsorbent can be obtained from the following formula:q_e_ = (C_0_ − Ce) V/m(1)

C_0_ is the initial concentration of the solution (mg L^−1^), Ce is the concentration when the adsorption equilibrium is reached) (mg L^−1^), m is the amount of adsorbent (g), and V is the volume of ionic solution (L).

A total of 10 mg of composite aerogel was immersed in 50 mL of 100 mg/L standard solution of C_d_^2+^, Cr^3+^, Co^2+^, Zn^2+^, and Pb^2+^, the mixture was then shaken in a constant temperature oscillator at 298 K 100 rpm for 120 min. The concentration of heavy metal ions in the supernatant was determined by ICP-MS.

The adsorption model is an important tool to describe the adsorption behavior and mechanism. In this study, two typical kinetic models, pseudo-first-order (Equation (1)) and pseudo-second-order models (Equation (2)), were used for fitting adsorption kinetic curves.

The pseudo-first-order model:(2)$$\ln\left(q_{e}-q_{t}\right)=\ln q_{e}-k_{1}t$$

The pseudo-second-order kinetic model:(3)$$\frac{t}{q_{t}}=\frac{1}{k_{2}q_{e}^{2}}+\frac{1}{q_{e}}t$$

The Langmuir isotherm model:(4)$$\frac{C_{e}}{q_{e}}=\frac{1}{q_{m}K_{L}}+\frac{C_{e}}{q_{m}}$$

The Freundlich isotherm model:(5)$$\ln q_{e}=\frac{1}{n}\ln Ce+\ln K_{F}$$
where *q_e_* (mg g^−1^) and *q_t_* (mg g^−1^) are the adsorption equilibrium and the adsorption amount at time t, respectively; k_1_ (min^−1^) and k_2_ (g mg^−1^ min^−1^) are the pseudo-first-order and pseudo-second-order rate constants, respectively; *C_e_* is the equilibrium concentration of metal icons in aqueous solution (mg∙L^−1^); *q_m_* and *K_L_* (L mg^−1^) are the Langmuir constants related to *q_e_* for a complete monolayer and energy of adsorption, respectively; *K_F_* (mg g^−1^) and n are the Freundlich constants that indicate the adsorption capacity and adsorption intensity, respectively.

After the adsorption reaches equilibrium, the aerogel adsorbing heavy metal ions was filtered out of the solution and washed several times with distilled water to remove any unadsorbed heavy metal ions. Then, the aerogel that reached the adsorption equilibrium was soaked in 50 mL of 0.1 M HCl, shaken at 298 K for 240 min, and regenerated in 0.1 M NaOH solution for 1 h. After that, it was washed to make it neutral, and cycled for several times to study the reuse of composite aerogel. The desorption efficiency (DE) was obtained from the following formula:DE = DI/AI × 100%(6)
where DI and AI is the amount of desorption ions and adsorption ions, respectively.

## 3. Results and Discussion

### 3.1. Morphology

As shown in Figure 1d, the nanocellulose had a diameter of 5~10 nm and a length of 10~20 μm. The nanocellulose with the ratio of height to diameter were interwoven to form a network structure. In addition, the C6 hydroxyl group of cellulose were oxidized to carboxyl groups, which introduced a negative charge on the cellulose surface, resulting in the nanocellulose dispersion colloids, which were extremely stable and did not precipitate for several months. Nanocellulose colloidal stability allowed it to evenly disperse graphene (Figure 1e illustration). The graphene appeared in irregular sheets with a diameter of about 50–70 μm (Figure 1e). SEM micrographs and digital pictures of the CA, CPA, and CPGA are shown in Figure 1a–c. CA was relatively brittle, and CA aerogels had relatively slender pores ranging from tens to hundreds of micrometers. The CPA and CPGA had small pore sizes owing to the partial melting of PVA in aerogel during heating esterification. Both CPA and CPGA had a rough surface with a soft and elastic structure. Moreover, the density of CPGA could be adjusted by adding different amounts of graphene from 10.00 to 20.00 mg cm^−3^.

The chemical composition and other physical properties of obtained aerogels were also characterized by FTIR. Figure 2a shows characteristic peaks of CA, CPA, and CPGA in the FTIR spectra. The characteristic peaks of CA showing at 3289, 2903, 1404, 1125, 1050, and 900 cm^−1^ were regarded as the typical cellulose peaks [21]. The peak at 3289 cm^−1^ was due to O−H stretching vibration absorption, 2903 cm^−1^ corresponded to the symmetric stretching vibration of C−H, and there was an asymmetric deformation vibration of −CH_3_ at 1404 cm^−1^. The bands at 1125, 1050, and 900 cm^−1^ were attributed to O−H bending and both symmetrical and asymmetrical stretching vibrations of C−O−C. The peak at 1639 cm^−1^ was assigned to the adsorbed water. CO−OH groups in CA was confirmed by the characteristic C=O stretching at 1795 cm^−1^ [22]. For CPA, the 1710 cm^−1^ ester carbonyl peak indicated that an esterification reaction took place between hydroxyl groups on cellulose/PVA and carboxyl groups of citric acid [23]. After the addition of graphene, CPGA did not show any new peaks, mainly because pure graphene did not have infrared absorption peaks [24].

The crystal structure of the aerogels was analyzed by XRD. Figure 2b shows the XRD patterns of CA, CPA, and CPGA. These three aerogels all had a peak at 21.87°, which could be ascribed to the characteristic peak of cellulose [25]. Additionally, the CA had a peak at 16.89°, which is also the characteristic peak of cellulose. CPA and CPGA did not have this peak; the reason for this could be that the reaction of cellulose and PVA produced ester bonds and partial hydrogen bonds, leading to the disappearance of the cellulose peaks. In CPGA, there was a characteristic peak of graphene at 24.6°, indicating that graphene was completely evenly dispersed in the composite aerogel [26].

To examine the thermal stability of the aerogels, the CA, CPA, and CPGA were studied by TGA. As revealed in Figure 2c, the TGA curves of CA can be divided into three stages [27]. The first stage was from room temperature to 200 °C, and the weight loss was relatively small (5%), which was ascribed to the water desorption of the cellulose. The second stage was from 200 to 320 °C; in this stage, the weight loss of CA rose sharply, which was attributed to removing of intramolecular bound water from cellulose (Figure 2d). The last stage indicated that the weight loss rate was relatively stable, which was ascribed to the carbonization of cellulose. As the temperature continued to rise, the cellulose macromolecular chains in CA began to break and form intermediates, which were then converted into small molecule substances such as CO, CO_2_, and H_2_O. After cross-linking and mixing with graphene, the maximum weight loss rate occurred at 160 °C, and then, the mass loss rate leveled off, mainly due to the poor thermal stability of PVA [28]. The side chain group of PVA started to decompose at 140 °C, mainly due to the elimination reaction of side chain hydroxyl in the molecule [29]. The thermogravimetric curves of CPA and CPGA composite aerogels were basically consistent, which was ascribed to the graphene with excellent thermal stability. The quality of the graphene did not change during the heating process. The residual carbon content of CPGA was relatively high, which can reach 41.2%.This was because the addition of graphene served as a physical barrier and promoted carbonization, thus improving the thermal performance of CPGA.

### 3.2. Mechanical Properties

As shown in Figure 3e,f, the aerogels possessed excellent compression performance both in air and underwater. When pressure was applied to the top of the aerogel, the aerogel was gradually flattened. After the pressure was removed, the aerogel could quickly return to its original height both in air and under water. As illustrated in Figure 3a, the aerogel was subjected to cyclic compression tests at a strains of 50% and 60%. The loading stress–strain curve can be roughly divided into three regions [30]. When the compressive strain was less than 28%, the stress increased linearly with the compressive stress, corresponding to the elastic region. When the compressive strain increased from 28 to 50, the stress increased linearly, corresponding to the collapse deformation region. The last densification region (strain > 50%) was marked by a rapid increase in compressive stress, owing to the densification of the pore. Furthermore, after 10 cycles compression at 50% strain, the stress–strain curve at 10th almost coincided with the 1st (Figure 3b), showing excellent fatigue-resistant property in air.

As an adsorption material, aerogels were mainly used in water or other organic liquids, and so, the mechanical properties of aerogels in water are very important [31]. As shown in Figure 3c, the aerogel was subjected to cyclic compression tests at strains of 60%, 70%, and 80% under the water. The aerogel can restore to its initial height even at 80% compression strain, indicating excellent underwater compression performance. Nevertheless, compared to compressing aerogel in air, the cyclic compression of aerogel under water had lower energy dissipation, mainly due to the water entering into the aerogel, which reduced the hydrogen bonding between the aerogel skeleton [26,31]. Similarly, the aerogel had excellent fatigue resistance under water (Figure 3c).

### 3.3. Adsorption Property

The heavy metal ions, including Cu^2+^, Cd^2+^, Pb^2+^, Cr^3+^, Co^2+^, and Zn^2+^, are common metallic contaminants in water. As shown in Figure 4a, the adsorption capacity of aerogels for the Cu^2+^, Cd^2+^, Pb^2+^, Cr^3+^, Co^2+^, and Zn^2+^ were 80.12 mg g^−1^, 102.23 mg g^−1^, 123.02 mg g^−1^, 62.38 mg g^−1^, 69.55 mg g^−1^, and 57.16 mg g^−1^, respectively. Here, Cu^2+^ was used as a cation adsorption model (Figure 4b) to investigate the static adsorption property. The adsorption capacity of Cu^2+^ for the CPA and CPGA was 50.78 mg g^−1^ and 84 mg g^−1^, respectively, which was better than that of the reported adsorption materials [32,33]. The adsorption capacity of CPGA was higher than that of CPA, mainly due to the high specific surface area and abundant surface functional groups of graphene [34]. Additionally, the CPA and CPGA showed a high adsorption rate less than 35 min, which may have been due to the high wettability and porosity of the aerogels.

As shown in Table 1 and Figure 4c,d, the values of R^2^ were generally higher than R^1^, indicating the pseudo-second-order model fitted the experimental data of all samples better compared to the pseudo-first-order model. This further illustrated that the adsorption process was mainly controlled by the chemical adsorption mechanism [35].

The desorption ability and cyclic adsorption performance of adsorbent are important indexes of adsorbent materials [36]. As illustrated in Figure 4e, the elution efficiency of the CPGA was about 11% in water, while over 80% in acid lotion including HNO_3_, H_2_SO_4_, and HCl. Finally, 1 M HCl was selected as the eluent and the cyclic adsorption capacity. As shown in Figure 4f, after five times of adsorption–desorption, the adsorption efficiency of CPGA decreased from 98.2 to 90.5, and the adsorption efficiency did not decrease significantly, indicating that the aerogels had excellent repeatability.

## 4. Conclusions

In conclusion, an originally robust CPGA was fabricated through PVA, graphene, and cellulose by a convenient super-assembly method which could be used as an efficient adsorbent of heavy metal ions in water. The covalent bond crosslinking formed in aerogels gave it excellent mechanical properties both in air and water, which can be compressed up to 80%. After 10 cycles compression at 50% strain, the stress–strain curve of CPGA at 10th almost coincided with the 1st, revealing outstanding fatigue resistance. Moreover, the adsorption capacity figures of aerogels for the Cu^2+^, Cd^2+^, Pb^2+^, Cr^3+^, Co^2+^, and Zn^2+^ were 80.12 mg g^−1^, 102.23 mg g^−1^, 123.02 mg g^−1^, 62.38 mg g^−1^, 69.55 mg g^−1^, and 57.16 mg g^−1^, respectively, showing high adsorption capacity. Additionally, the CPGA revealed a high adsorption rate less than 35 min. The pseudo-second-order kinetic model fitted better than the pseudo-first-order kinetic model in describing the adsorption process of Cu^2+^ on CPGA. The Freundlich isotherm was confirmed as a more suitable model to describe the adsorption equilibrium of CPGA than the Langmuir isotherm model. Thus, such a compressible and durable CPGA is a very promising material to adsorb heavy metal ions in industrial wastewater.

## Figures and Tables

**Figure 1 polymers-15-02434-f001:**
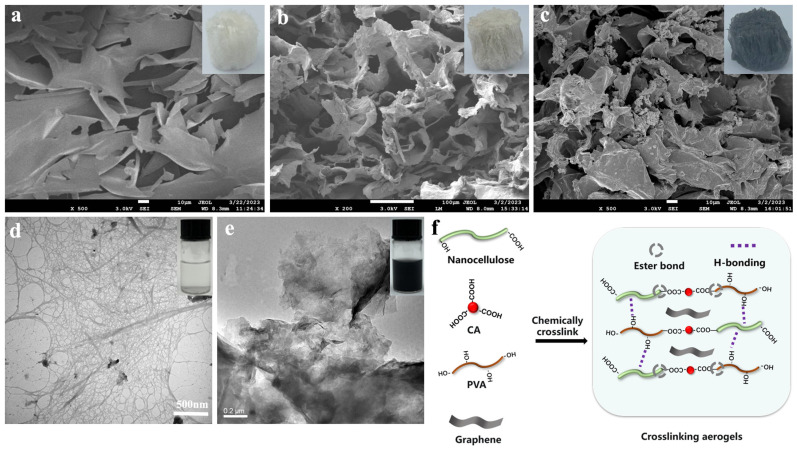
SEM images and digital pictures of (**a**) CA, (**b**) CPA, and (**c**) CPGA, respectively. TEM images and digital pictures of (**d**) TOC and (**e**) graphene, respectively. (**f**) Schematic illustration of the crosslinking mechanism of the CPGA.

**Figure 2 polymers-15-02434-f002:**
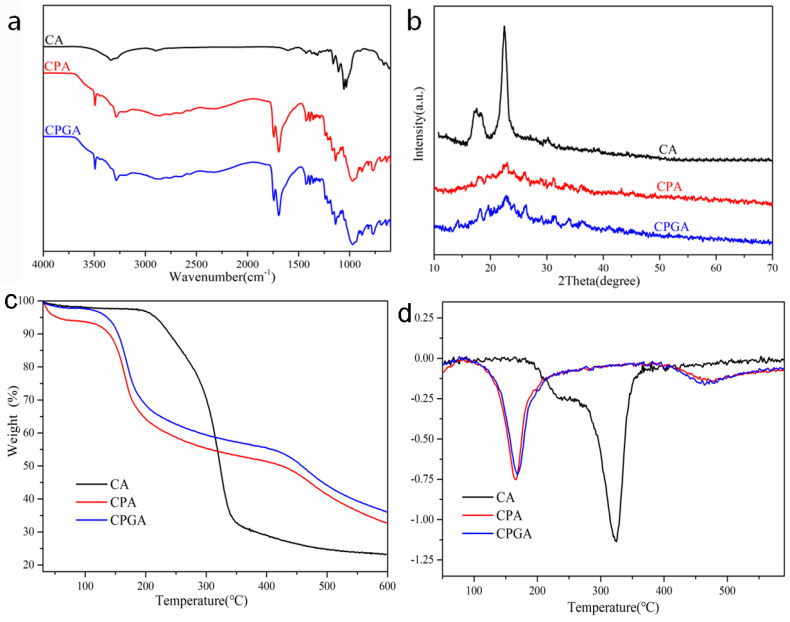
(**a**) FTIR spectra, (**b**) XRD patterns, (**c**) TGA, and (**d**) DTG curves of CA, CPA, and CPGA, respectively.

**Figure 3 polymers-15-02434-f003:**
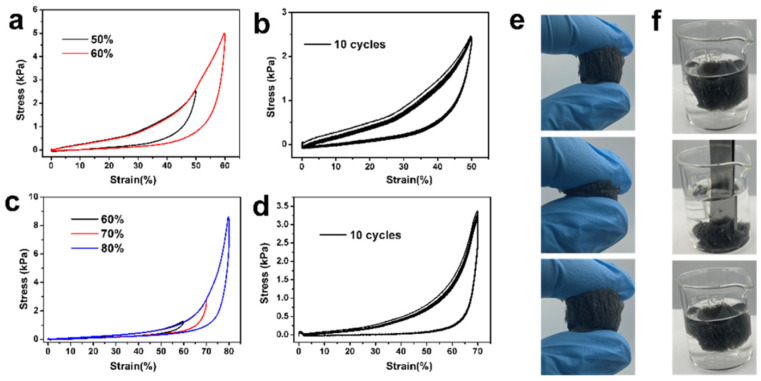
(**a**) Stress–strain curves of aerogel at 50% and 60%stains in the air. (**b**) Stress–strain curves of aerogel at 10th cycles at 50% strain in the air. (**c**) Stress–strain curves of aerogel under different stains under the water. (**d**) Stress–strain curves of aerogel at 10th cycles at 50% strain under the water. Photo of aerogel bouncing back under compression in air (**e**) and underwater (**f**), respectively.

**Figure 4 polymers-15-02434-f004:**
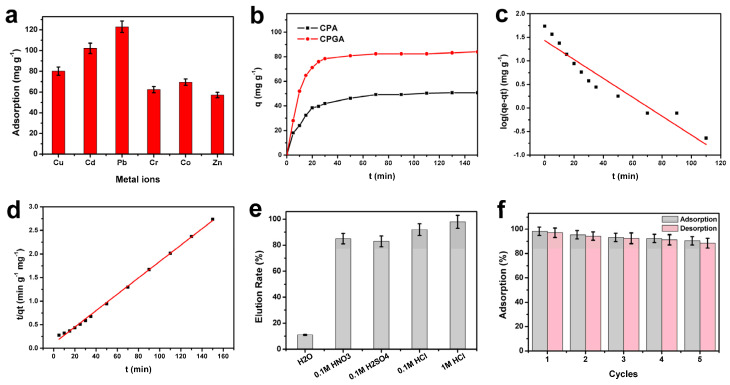
(**a**) Adsorption capacities of CPGA with various metal ions. (**b**) The adsorption efficiency as a function of contact time for CPA and CPGA. (**c**) Pseudo-first-order kinetics and pseudo-second-order kinetics (**d**) fit of Cu^2+^ adsorption on CPGA. (**e**) The elution efficiency of the CPGA. (**f**) The impact of recycling on adsorption capacity.

**Table 1 polymers-15-02434-t001:** Calculated constants of pseudo-first-order and pseudo-second-order models for Cu^2+^ adsorption on CPGA.

Metal Ion	q_e_ (exp.) mg/g	k_1_ (min^−1^)	R^1^	k_2_ (g mg^−1^ min^−1^)	R^2^
Cu^2+^	80.12	−0.02088	0.87514	0.0078	0.99812

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
