# Peer review of "Covalent Crosslinking Cellulose/Graphene Aerogels with High Elasticity and Adsorbability for Heavy Metal Ions Adsorption"

_polymers, 2023, doi:10.3390/polym15112434_

Round 1
Reviewer 1 Report
Well written article.
The main criticism concerns
1- The large-scale implementation of these materials
2- Their incorporation into water purification devices
3- Recycling these materials at the end of their life
Do the authors have any answers to add to the text?
In addition, the legends of the figures must be checked
Perfectly understandable paper
Author Response
Do the authors have any answers to add to the text?
Response: Thanks for your kind review. Your summary is very comprehensive. We have not any answers to add to the text.
In addition, the legends of the figures must be checked
Response: Thanks for your reminder. We have checked all figures in the update manuscript.
Reviewer 2 Report
The manuscript reports the development of aerogels based on covalent crosslinking cellulose nanofibers containing polyvinylalcohol and graphene for heavy metal removal. The aerogels were characterized by X-ray diffraction (XRD), thermal and mechanical analysis as well as by FTIR spectroscopy and scanning electron microscopy. In addition, the adsorption capacity of the aerogels versus different metals was studied.
Some experimental details are missing from the manuscript. Furthermore, the authors did not discuss the obtained results well. The English should be revised.
The authors should consider the comments pointed out below:
Line 101. The acronym CA for citric acid should be dropped as the same acronym has been given to TOC aerogel (line 138). Moreover, it doesn’t seem necessary.
Line 153. As for the definition of C0 and Ce parameters, what does initial solubility of the ionic solution and ionic solubility at equilibrium mean? Shouldn't the parameters correspond to the initial and equilibrium concentration, respectively?
Line 156. How was the aerogel amount put in contact with the solution containing the metal ions chosen? Have the authors experimented different amounts of aerogel? Also, as the pH value can influence the adsorption of heavy metal ions, at what pH were the absorption experiments carried out?
Line 167. Definition of desorption efficiency. Could the authors explain what they mean by desorption number of Cu2+ and absorption number of Cu2+?
Lines 206-207. FTIR analysis. The absorption peaks of CA aerogel reported in the text are not all visible. Additionally, assignments of the peaks should be provided. Why is the C=O carboxyl peak not visible at all in the CA sample?
Lines 207-209. The sentence should be corrected. The esterification reaction took place between hydroxyl groups of PVA and carboxyl groups of CA.
Lines 223-225. What is the weight loss in the range 200-320°C attributed to? Also, the authors should provide more explanations on the thermal stability of CPA and CPGA samples lower than that of CA. What is the influence of PVA and graphene on the aerogel structure? Why does the CPGA sample containing graphene behave like CPA?
Lines 272-274. Why were not measurements of the aerogel swelling performed?
Lines 276-278. Please, complete the caption of the figure 4 (describe what figures 4e and 4f are).
Lines 299-301. In the conclusion the authors state that the Freundlich isotherm is the most suitable model to describe the adsorption equilibrium of CPGA. The study of the Freundlich and Langmuir isotherms and the related data are missing from the manuscript.
The reference 34 is not present in the text.
The English should be revised.
Author Response
Review reports 2:
The manuscript reports the development of aerogels based on covalent crosslinking cellulose nanofibers containing polyvinylalcohol and graphene for heavy metal removal. The aerogels were characterized by X-ray diffraction (XRD), thermal and mechanical analysis as well as by FTIR spectroscopy and scanning electron microscopy. In addition, the adsorption capacity of the aerogels versus different metals was studied.
Some experimental details are missing from the manuscript. Furthermore, the authors did not discuss the obtained results well. The English should be revised.
The authors should consider the comments pointed out below:
Line 101. The acronym CA for citric acid should be dropped as the same acronym has been given to TOC aerogel (line 138). Moreover, it doesn’t seem necessary.
Response: Thanks! We have deleted CA.
Line 153. As for the definition of C0 and Ce parameters, what does initial solubility of the ionic solution and ionic solubility at equilibrium mean? Shouldn't the parameters correspond to the initial and equilibrium concentration, respectively?
Response: Thank you for kind review. C0 (mg L-1) is the initial concentration of the solution, and Ce (mg L-1) is the concentration when the adsorption equilibrium is reached.
Line 156. How was the aerogel amount put in contact with the solution containing the metal ions chosen? Have the authors experimented different amounts of aerogel? Also, as the pH value can influence the adsorption of heavy metal ions, at what pH were the absorption experiments carried out?
Response: Thanks! That's a very good question. The amount of aerogel put in the solution was estimated by several tests. If less aerogel was added to the solution, the concentration of ion change in solution is difficult to measure accurately. If the amount of aerogel added to the solution is too much, it is difficult to reach adsorption saturation. Finally, about 0.01g aerogel was selected for adsorption experiment. The pH value of the solution is a key parameter for evaluating the adsorption capacity of adsorption materials. The pH of the solution was measured to be 5.0 ± 0.5.
Line 167. Definition of desorption efficiency. Could the authors explain what they mean by desorption number of Cu2+ and absorption number of Cu2+?
Response: Thanks! Desorption efficiency refers to the percentage of the amount of desorbed substance to the total amount of adsorbed substance to be measured on the solid adsorbent. Desorption efficiency (DE) =desorbed substance/ adsorption subastance *100%. The desorption number of Cu2+ is inaccurate and has been changed.
Lines 206-207. FTIR analysis. The absorption peaks of CA aerogel reported in the text are not all visible. Additionally, assignments of the peaks should be provided. Why is the C=O carboxyl peak not visible at all in the CA sample?
Response: Thanks! The chemical composition and other physical properties of obtained aerogels were also characterized by FTIR. Fig.2a shows characteristic peaks of CA, CPA and CPGA in the FTIR spectra. The characteristic peaks of CA showing at 3289, 2903, 1404, 1125, 1050 and 900 cm-1 are regarded as the typical cellulose peaks[1]. The peak at 3289 cm−1 due to O-H stretching vibration absorption, 2903cm-1 corresponds to the symmetric stretching vibration of C-H and there is an asymmetric deformation vibration of - CH3 at 1404 cm-1. The bands at 1125, 1050 and 900 cm−1 were attributed to O-H bending and both symmetrical and asymmetrical stretching vibrations of C-O-C. The peak at 1639 cm−1 was assigned to the adsorbed water. CO-OH groups in CA is confirmed by the characteristic C=O stretching at 1795 cm−1.[2] Due to the small amount of carboxyl group of CA, there is only very weak vibration in the infrared image. For CPA, the 1710 cm-1 ester carbonyl peak indicates that an esterification reaction took place between hydroxyl groups on cellulose and carboxyl groups of CA.[3] After the addition of graphene, CPGA did not show any new peaks, mainly because pure graphene did not have infrared absorption peaks.[4]
Lines 207-209. The sentence should be corrected. The esterification reaction took place between hydroxyl groups of PVA and carboxyl groups of CA.
Response: Thank you for your reminder, it has been modified. Line 207-209 The esterification reaction took place between hydroxyl groups of cellulose/PVA and carboxyl groups of citric acid.
Lines 223-225. What is the weight loss in the range 200-320°C attributed to? Also, the authors should provide more explanations on the thermal stability of CPA and CPGA samples lower than that of CA. What is the influence of PVA and graphene on the aerogel structure? Why does the CPGA sample containing graphene behave like CPA?
Response: Thanks. The first stage is from room temperature to 200 °C, and the weight loss is relatively small (5%), which is ascribed to the water desorption of the cellulose. The second stage is from 200 to 320 °C; in this stage, the weight loss of CA rises sharply, which is attributed to removing of intramolecular bound water from cellulose (Fig. 2d). The last stage indicates that the weight loss rate is relatively stable, which is ascribed to the carbonization of cellulose. As the temperature continues to rise, the cellulose macromolecular chains in CA began to break and form intermediates, which are then converted into small molecule substances such as CO, CO2, and H2O. After cross-linking and mixing with graphene, the maximum weight loss rate occurred at 160 °C, and then the mass loss rate is leveling off, mainly due to the poor thermal stability of PVA.[5]. The side chain group of PVA starts to decompose at 140 ℃, mainly due to the elimination reaction of side chain hydroxyl in the molecule.[6] The thermogravimetric curves of CPA and CPGA composite hydrogels are basically consistent, which is ascribed to the graphite with excellent thermal stability. The quality of the graphene does not change during the heating process. The residual carbon content of CPGA is relatively high, which can reach 41.2%.This is because the addition of graphene serves as a physical barrier and promotes carbonization, so then improving the thermal performance of CPGA.
Lines 272-274. Why were not measurements of the aerogel swelling performed?
Response: This is a good question. The CPGA contains a large number of robust covalent bonds, making it extreme stability in water without any swelling. Hence, the swelling performance of aerogel data is not list in the article.
Lines 276-278. Please, complete the caption of the figure 4 (describe what figures 4e and 4f are).
Response: Thanks. We have added the description of Figure 4. (e) the elution efficiency of the CPGA (f) The impact of recycling on adsorption capacity
.
Lines 299-301. In the conclusion the authors state that the Freundlich isotherm is the most suitable model to describe the adsorption equilibrium of CPGA. The study of the Freundlich and Langmuir isotherms and the related data are missing from the manuscript.
Response: Thanks for you kind remind. The study of the Freundlich and Langmuir isotherms and the related date are listed in Table 1. By computation, values of R2 are generally higher than R1, indicating the pseudo-second-order model fitted the experimental data of all samples better compared to the pseudo-first-order model.
The reference 34 is not present in the text.
Response: Thanks. The reference 34 has been deleted.

Reviewer 3 Report
The Authors should indicate te salts used for preparation of Cd2+, Cr3+, Co2+, Zn2+ and Pb2+ solutions.
Int the line 179 the expression "mental icons" should be corrected.
In the Figure 1f the symbol of cellulose is missing.
The FTIR analysis should be described more thoroughly indicating all wisible bands in the spectra.
The thermal degradation of composites is not properly derscribed. Why there is such difference between composites and pure cellulose?
In the Figure 4 error bars should be inserted.
In the line 286 and 289 inappropriate figure is mentioned.
The Conclusions should cover all the results obtained.
Some language issues should be corrected (shovn above).
Author Response
The Authors should indicate te salts used for preparation of Cd2+, Cr3+, Co2+, Zn2+ and Pb2+ solutions.
Response: Thanks. Metal element(Cd2+, Cr3+, Co2+, Zn2+ and Pb2+ solution) standard stock solution (100 μ G • m L-1) were prepared by CdCl2·2.5H2O, CrCl3·6H2O, CoCl2, ZnCl2, and PbCl2 , respectively.
Int the line 179 the expression "mental icons" should be corrected.
Response: "metal icons” has been corrected.
In the Figure 1f the symbol of cellulose is missing.
Response: Figure 1f the symbol of cellulose has been added.
The FTIR analysis should be described more thoroughly indicating all wisible bands in the spectra.
Response: Thanks! FTIR analysis has been described more thoroughly and added in article.
The thermal degradation of composites is not properly derscribed. Why there is such difference between composites and pure cellulose?
Response: The TGA of composites has been derscribed properly in the updated manuscript .Because the PVA that is not fully involved in the reaction is prone to water absorption and has a low decomposition temperature in the early stage.
In the Figure 4 error bars should be inserted.
Response: Thanks. The error bars of Fig 4 had been added.
In the line 286 and 289 inappropriate figure is mentioned.
Response: Thanks. It has been changed.
The Conclusions should cover all the results obtained
Response: Thanks. We have revised the conclusion and marked with yellow.

Round 2
Reviewer 2 Report
I appreciate the efforts made by the authors to improve the manuscript. They accepted all suggestions made by the reviewer.